# Extraction of Acetogenins Using Thermosonication-Assisted Extraction from *Annona muricata* Seeds and Their Antifungal Activity

**DOI:** 10.3390/molecules27186045

**Published:** 2022-09-16

**Authors:** Brandon Alexis López-Romero, Gabriel Luna-Bárcenas, María de Lourdes García-Magaña, Luis Miguel Anaya-Esparza, Luis Gerardo Zepeda-Vallejo, Ulises Miguel López-García, Rosa Isela Ortiz-Basurto, Gabriela Aguilar-Hernández, Alejandro Pérez-Larios, Efigenia Montalvo-González

**Affiliations:** 1Laboratorio Integral de Investigación en Alimentos, Tecnológico Nacional de México/Instituto Tecnológico de Tepic, Av. Tecnológico 2595, Lagos del Country, Tepic 63175, Nayarit, Mexico; 2Centro de Investigación y de Estudios Avanzados, Libramiento Norponiente 2000, Fracc. Real de Juriquilla, Santiago de Querétaro, Querétaro 76230, Mexico; 3División de Ciencias Agropecuarias e Ingenierías, Centro Universitario de los Altos, Universidad de Guadalajara, Av. Rafael Casillas Aceves 1200, Guadalajara 47600, Jalisco, Mexico; 4Departamento de Química Orgánica, Escuela Nacional de Ciencias Biológicas, Instituto Politecnico Nacional, Av. Prolongación de Carpio y Plan de Ayala s/n, Col. Santo Tomás, Delegación Miguel Hidalgo, Ciudad de Mexico 11340, Mexico

**Keywords:** *A. muricata* seeds, acetogenins, thermosonication-assisted extraction

## Abstract

The objective of this work was to find the optimal conditions by thermosonication-assisted extraction (TSAE) of the total acetogenin content (TAC) and yield from *A. muricata* seeds, assessing the effect of the temperature (40, 50, and 60 °C), sonication amplitude (80, 90, and 100%), and pulse-cycle (0.5, 0.7, and 1 s). In addition, optimal TSAE conditions of acetogenins (ACGs) were compared with extraction by ultrasound at 25 °C and the soxhlet method measuring TAC and antioxidant capacity. Moreover, solubility and identification of isolated ACGs were performed. Furthermore, the antifungal activity of ACGs crude extract and isolated ACGs was evaluated. Optimal TSAE conditions to extract the highest TAC (35.89 mg/g) and yield (3.6%) were 50 °C, 100% amplitude, and 0.5 s pulse-cycle. TSAE was 2.17-fold and 15.60-fold more effective than ultrasound at 25 °C and the Soxhlet method to extract ACGs with antioxidant capacity. Isolated ACGs were mostly soluble in acetone and methanol. Seven ACGs were identified, and pseudoannonacin was the most abundant. The inhibition of *Candida albicans, Candida krusei, and Candida tropicalis* was higher from isolated ACGs than crude extract. TSAE was effective to increase the yield in the ACGs extraction from *A. muricata* seeds and these ACGs have important antifungal activity.

## 1. Introduction

Annonaceae acetogenins (ACGs) are functional aliphatic molecules of 35 to 37 carbons, bearing a terminal γ-lactone, a variable number of attached hydroxyl groups, and one, two, or three tetrahydrofuran rings in their central region or other functional groups. These compounds are unique to the Annonaceae family and are widely studied in ethnobotany because they present biological activities such as antimicrobial, antiviral, and antitumoral [1].

ACGs are found in the roots, stem-bark, leaves, pulp, peel, and seeds from Annonanaceae plants. Seeds from *A. muricata* contain the highest concentration of ACGs. Seeds represent 5–10% of the whole fruit and are considered a waste; therefore, they can be used as a rich source of ACGs [2]. However, until now, the yield of ACGs by conventional extraction methods is low. Using maceration was obtained 1% ACGs [3]; while using the Soxhlet method was extracted 0.04–0.1% ACGs [4], and with supercritical fluids, 1.68–2.09 mg ACGs/g [5,6], all from *A. muricata* seeds. Nonetheless, ultrasound-assisted extraction (UAE) is a current technology used to extract ACGs from *A. muricata* pulp and by-products, including seeds [7]. The optimization of the UAE conditions at 25 °C to obtain an ACGs crude extract from *A. muricata* seeds was developed by Aguilar-Hernández et al. [7]. The authors found that the optimal UAE conditions to get a 1.3% yield of ACGs crude extract from whole seeds were 15 min of extraction time, 0.7 s pulse-cycle, and 100% sonication amplitude at 25 °C, using chloroform as solvent. However, the yield of ACGs extraction could increase if the thermosonication conditions are proven.

Combining ultrasound and heat to extract metabolites from natural sources is called thermosonication-assisted extraction (TSAE). It is a promising alternative for removing thermostable bioactive compounds from complex matrices at temperatures from 28 °C to 60 °C [8]. The ultrasonic waves and heat increase the yield of compounds from plant tissues because heat causes the growth and rapid implosion of microbubbles, increasing the diffusivity of the solvent in the matrix [9]. Physical changes more frequently reported with thermosonication are fragmentation, detexturation (disruption), erosion, and sonoporation with damage to cell membranes and the internal distortion of the organelles that causes the release of substances from the solid phase to the solvent [9,10]. Agcam et al. [8] reported that TSAE maximized the yield of five different anthocyanins from black carrot pomace using the extraction conditions at 183.1 W/g, 50 °C, and 20 min. On the other hand, López-Ordaz et al. [11] demonstrated that the optimal TSAE conditions to increase the oil yield (61.12%) from the seeds of *Ricinus communis L* were 50% sonication amplitude, 35 min, and a solid:liquid ratio of 1:10 (w:v), compared with Soxhlet extraction since the yield was 57.3% after 8 h of extraction. Moreover, the extraction of peanut oil was investigated using TSAE. The optimum TSAE conditions were 4 min, 60 °C, a ratio solvent: solid of 9:l (*v*/*w*), and 100% sonication amplitude with a maximum extraction yield of 39.86% [12]. 

On the other hand, it has been demonstrated that *A. muricata* extracts exhibit antifungal activity. Rizwana et al. [13] evaluated the in vitro and in vivo effect of extracts from the pulp and seeds of *A. muricata* at different concentrations (0.5, 1, 1, 2, 2, 4, 6% *w*/*v*) as an alternative to synthetic fungicides against *Alternaria alternata* causal agent of tomato fruit black spot. These authors reported that methanol extracts of seeds (6% *w*/*v*) were more potent in inhibiting *A. alternata* than pulp extracts. The in vitro assay showed maximum inhibition of mycelial growth of *A. alternata* (90%) and a marked reduction in lesion diameter (2.1 mm) in the in vivo assay with 84% disease inhibition in fruit treated with seed extracts. Likewise, León-Fernández et al. [14] tested the effect of 15 fractionated extracts (chloroform: methanol) of soursop pulp against *Colletotrichum gloeosporioides* and *Rhizopus stolonifer*, finding that three rich ACGs fractions were the most effective against *C. gloeosporioides* and *R. stolonifer* with 59% and 38% inhibition, respectively. Until now, there is no information on the application of TSAE in the extraction of ACGs from *Annona* seeds nor the antifungal activity of isolated ACGs. 

The objective of this work was to extract ACGs from *A. muricata* seeds using the TSAE and compare the optimal TSAE conditions with ultrasound-assisted extraction (UAE) at 25 °C and the Soxhlet method measuring total ACGs and their antioxidant capacity. In addition, ACGs crude extract was purified (named isolated ACGs) and analyzed by HPLC-DAD. Moreover, the antifungal activity of ACGs crude extract or isolated ACGs was evaluated.

## 2. Results and Discussion

### 2.1. Total Acetogenin Content (TAC) and Yield from the Defatted Endosperm of A. muricata Seeds by Thermosonication-Assisted Extraction (TSAE)

TAC and yield are shown in Table 1. There were significant differences (*p* < 0.05) between treatments. TAC and yield depend on each factor and their interactions. The values ranged from 27.25 to 34.33 mg/g DW and 2.73 to 3.44% yield, obtaining the highest TAC and yield (34.35 mg/g DW, 3.44%) at X_ET_ 50 °C, X_SA_ 100%, X_PC_ 0.5 s. In contrast, the conditions of X_TE_ 50 °C, X_SA_ 100%, and X_PC_ 1 s presented the lowest TAC and yield (27.25 mg/g DW, 2.73%).

The increasing temperature in the ultrasonic medium decreases the implosion threshold of the bubbles in the cavitation. Thus, a larger number of bubbles will implode in a short period of time causing considerable damage to the cell wall or an increase in cell pore size. It improves the diffusivity of the solvent within the matrix, increasing the yield; however, the effect depends on the TSAE conditions [15,16]. In this study, the increase in the yield of ACGs is attributed to the previous preparation of the raw material (defatted) and the synergistic effect between temperature (50 °C), sonication amplitude (100%), and pulsed sonication (0.5 s pulse-cycle). Nonetheless, if the extraction temperature is 40 ± 2 °C, it is insufficient to increase the yield independently of sonication amplitude and pulse-cycle. However, if the extraction temperature is 60 ± 2 °C, 100% sonication amplitude, and 1 s pulse-cycle, there is an excess of energy in the medium, which degrades ACGs [17]. According to Neske et al. [18], the structures of ACGs are liable to change above 60 °C.

### 2.2. Optimal TSAE Conditions to Extract Total Acetogenins and Yield from the Defatted Endosperm of Annona muricata Seeds

Response surface methodology (RSM) of multiple regression was performed with TAC and yield data as a function of temperature, pulse-cycle, and sonication amplitude. The second-order polynomial equations (Table 2) describe the individual and combined effects of all the independent variables on TAC and yield. According to RSM, the TAC and yield values were predicted using mathematical models (Equations (1) and (2) at a 95% confidence level (See Table 2).

The individual effects of each variable and their interactions on the TAC and yield are presented in Table 3. The analysis of variance (ANOVA) of quadratic polynomial models demonstrated a significant effect of variables by F-value and *p*-value. The sonication amplitude showed an effect on the linear terms (*p* < 0.001), while the extraction temperature, sonication amplitude, and pulse-cycle and their interactions had significance for the quadratic terms (*p* < 0.05), which demonstrate that TAC and yield depended on these operation variables. Moreover, ANOVA showed that experimental data of TAC and yield presented a high correlation (R^2^ = 0.98 and R-adjust = 0.97) and an excellent fit to the models (lack of fit, *p* > 0.05). The lack of fit showed the model fitness to indicate an approximation to a real system; also, regression coefficients were significant. The RSM using Box–Behnken design with the variables such as sonication amplitude, pulse-cycle, and extraction time to obtain optimal UAE conditions of TAC has been reported with *A. muricata* seeds [7] and oil from *Annona squamosa* seeds [19].

The authors found similar correlation coefficients (R^2^ = 0.97 and R^2^ = 0.99, respectively) with this experiment. They concluded that RSM is an efficient tool to optimize extraction conditions of bioactive compounds from these Annonaceous plants. No comparable data to this experiment were reported by Aguilar-Hernández et al. [7] during UAE (24 kHz, 100% amplitude, 15 min, 0.7 pulse-cycle) of TAC (13.01 mg/g) from whole *A. muricata* seeds. It is attributed to the heat and ultrasound combination, increased extraction time, tegument elimination, methanol as solvent, and defatted endosperm.

The response surface plots (Figure 1 and Figure 2) show the effect of significant interactions (*p* < 0.05) between different TSAE conditions on the TAC and the yield, varying the ultrasonic amplitude. The elliptical shape of the plots shows the interactions between the corresponding variables. At 90% and 100% sonication amplitude (Figure 1A,B and Figure 2A,B), the highest TAC and yield were obtained with 0.5 s pulse-cycle and 50 °C, while at 80% amplitude, high temperature (60 °C) and pulse-cycle (1 s) are required for the highest TAC and yield (Figure 1C and Figure 2C). Pareto diagrams (Figure 1D and Figure 2D) exhibited a synergistic or antagonistic effect of independent variables and their interactions on the TAC and yield at a 95% confidence level. The principal results of linear and quadratic parameters were X_ET_^2^*X_PC_ > X_PC_ > X_PC_^2^.

The pulsed or constant ultrasonic energy and the extraction temperature are the most critical factors in extracting lipophilic compounds by TSAE [12,20]. However, the high temperature, high sonication amplitude or acoustic energy, constant acoustic irradiation (1 s pulse-cycle), and long extraction times may degrade these bioactive compounds due to the cavitation heat and mechanical stress [20]. In this study, the TAC was lowest when the TSAE was applied to a 1 s pulse-cycle, 50 °C, and 100% sonication amplitude. Agcam et al. [8] reported that the highest ultrasound energy density (400 J/g) caused the lowest anthocyanins content from black carrot pomace compared to 183 J/g. Similarly, the highest sonication amplitude (75%) did not extract more oil from the seeds of *Ricinus communis* compared to the 50% sonication amplitude [11]. When the amplitude of the ultrasonic waves is high, it increases the pressure in the extraction system. It can induce instability of the wave in the liquid medium, causing the transfer phenomena to be transient [21].

On the other hand, the viscosity of the medium may also change during the extraction time. This change affects cavitation by a decrease in energy transmission, which in turn decreases the yield [11]. It agrees with other reports where amplitude in UAE on the extraction of polyphenols, acetogenins, and alkaloids from *A. muricata* was demonstrated [7,22,23]. Chemat et al. [9] recommend that TSAE is used with high amplitudes when the processing is applied to viscous liquids, but the energy to reach the cavitation threshold must be considered. Therefore, optimization studies to operate TSAE should be carried out for each study matrix. 

Thus, the optimal TSAE conditions to extract ACGs from the defatted endosperm of *A. muricata* seeds are shown in Table 4. The optimal conditions were X_TE_ 50 °C; X_SA_ 100% and X_PC_ 0.5 s. The predicted optimal response was 33.98 mg of TAC/g DW and a 3.43% yield.

### 2.3. Model Reliability and Comparison of Optimal TSAE Conditions with Ultrasound and Soxhlet Method to Extract Total Acetogenins from the Defatted Endosperm

The experimental validation of the optimal TSAE conditions exhibited 35.89 mg/g of ACGs and a 3.60% yield (Table 5). This result is within the confidence limits obtained with the predicted values (See Table 4); therefore, model reliability was demonstrated.

On the other hand, Table 5 shows that TAC by TSAE was higher 2.17-fold and 15.60-fold than UAE and Soxhlet methods. Similar results were reported by López-Ordaz et al. [11] using TSAE (50% sonication amplitude, 30 min, 130 W) of castor oil (61.2% yield) compared with the Soxhlet method for 8 h (57.3% yield). Urango et al. [20] mentioned that TSAE showed advantages compared to Soxhlet by longer extraction time and degradation of bioactive compounds.

Moreover, acetogenic extract exhibited the highest AOX by the different assays as follows ABTS (4308.09 μmol/g DW) > DPPH (1668.29 μmol/g DW) > FRAP (1512.89 μmol/g DW). AOX has been reported in acetogenic extracts from *Annona cornifolia* seeds [24] and *A. muricata* pulp [25]. Our results coincided with León-Fernandez et al. [25], who reported a greater antioxidant activity by ABTS assay than DPPH assay in an ACGs crude extract and ACGs semi-purified fractions from soursop pulp. The AOX activity of acetogenins is attributed to the α, ß-unsaturated lactone ring with allylic hydrogens. Acetogenins donate hydrogens to free radicals, and they are stabilized through electron delocalization in the α, ß-unsaturated lactone ring [24,25].

### 2.4. Solubility and Analysis by HPLC-DAD of Isolated Acetogenins

These are the first reported results of solubility for isolated acetogenins. The high solubility of isolated ACGS was observed in polar protic solvents such as methanol and ethanol (Figure 3). The hydroxyl group of ethanol and methanol can interact with hydroxyl groups of ACGs, causing their dissolution. However, ACGs are insoluble in water (polar protic solvent) due to insufficient hydroxyl groups to form hydrogen bonding [26].

In addition, ACGs exhibited solubility in polar aprotic solvents such as acetone > ethyl acetate > dichloromethane. It was because the polar aprotic solvents are proton acceptors that can dissolve charged compounds forming covalent bonds [27]. Nonetheless, ACGs were insoluble in acetonitrile (polar aprotic solvent with high dielectric constant), attributed to a low proportion of -OH in ACGs structure to interact with the nitrogen atom of acetonitrile. Similarly, isolated ACGs were insoluble in non-polar solvents (petroleum ether and hexane).

The presence of two acetogenins (pseudoannonacin and annonacin) was demonstrated from isolated ACGs of *A. muricata* seeds (see Figure 4A,B). To our knowledge, pseudoannonacin has not been reported in *A. muricata* seeds. In addition, the presence of bullatacin, squamostatin-D, squamocin, isodesacetyluvaricin, and desacetyluvarin is possible. It is proposed because we evaluated the same chromatographical separation conditions of ACGs samples by HPLC-DAD that Yang et al. [5,6] used. These authors reported the exact retention times of ACGs that we found in this experiment (Figure 4B). Moreover, they concluded that these isolated ACGs are present in *A. muricata* seeds.

Moreover, at least three unknown ACGs were detected in this work. However, individual separation of ACGs is necessary to submit them to spectroscopic analysis (NMR and LC-MS/MS) and to know the absolute chemical structure of isolated ACGs; however, this analysis will be performed in future investigations.

On the other hand, peak separation improved when the column temperature was reduced during elution at 25 °C (Figure 4C), although retention times were prolonged. In addition, the best peak separation was observed when column temperature was maintained at 20 °C (Figure 4D). The solubility of the compounds increases with temperature [27]. Thus, when the column temperature was at 30 °C, the ACGs were mainly soluble in the mobile phase, decreasing their retention in the stationary phase. In contrast, by reducing the column temperature to 20 °C, the solubility of ACGs in the mobile phase decreased, leading to higher retention in the stationary phase and better separation. However, when the column temperature was 15 °C or 10 °C (data not shown), the separation of ACGs was unsuccessful. These results are essential for the future separation of individual ACGs. 

Table 6 shows that pseudoannonacin (60.22% area) was mostly concentrated followed by desacetyluvaricin (8.1% area), annonacin (6.17% area), unknown ACG_1_ (6.09% area), squamostatin-D (4.36% area), isodesacetyluvaricin (3.45% area), unknown ACG_3_ (3.37% area), unknown ACG_2_ (3.09% area), squamocin (2.99% area), and bullatacin (2.16% area). Approximately fifty-two different ACGs have been identified from *A. muricata* seeds [6,18], including those found in this work; however, the presence or not of some of them depends on the extraction solvent, extraction method, and purification process [18]. 

In this work, pseudoannonacin (350 mg/g) and annonacin (15 mg/g) (Figure 5) were quantified. The total content of ten isolated ACGs by supercritical fluid CO2 extraction from *A. muricata* seeds was 1.69 to 2.09 mg/g, according to Yang et al. [5,6]. These authors did not report pseudoannonacin and annonacin. Furthermore, Wurangian [4] and Ranisaharivony et al. [28] found 0.45 mg/g and 5.64 mg/g of annonacin extracted from *A. muricata* seeds, respectively, using ethanol as solvent and Soxhlet as an extraction method. Therefore, it can be inferred that TSAE is a technology more efficient than maceration, Soxhlet, and ultrasound at 25 °C to obtain a significant yield of isolated ACGs from *A. muricata* seeds.

### 2.5. Antifungal Activity of Acetogenin Crude Extract and Isolated Acetogenins

Table 7 shows the effect of the crude extract of ACGs and isolated ACGs against *C. albicans, C. krusei, C. tropicalis, and C. glabrata*. Significant statistical differences (*p* < 0.05) were observed between the different concentrations, and the antifungal effect varied depending on the *Candida* species. The most significant inhibition of isolated ACGs at 800 µg/mL was *C. albicans* (15.50 mm) > *C. tropicalis* (14 mm) > *C. krusei* (13.50 mm) > *C. glabrata* (8.50 mm). Moreover, it was evident that isolated ACGs (400 µg/mL) had equal or higher inhibition zone in *C. albicans*, *C. krusei,* and *C. tropicals* than ketoconazole (500 µg/mL). Rustanti and Fatmawati [29] evaluated the effect of fractionated extracts of *A. muricata* leaves (65–149 mg/mL) against *C. albicans*, finding an inhibition zone of 10–23 mm. Therefore, we can infer that isolated ACGs were most active. ACGs inhibit the NADH-ubiquinone oxide-reductase (Complex I) and NADH ubiquinone oxidase enzymes in the mitochondrial membrane. This inhibition decreases ATP production, causing cellular apoptosis. Fungi and yeasts are organisms with NADH-ubiquinone-6-oxidoreductase type II enzymes or NADH dehydrogenases; thus, ACGs can inhibit the action of these enzymes and consequently promote cell apoptosis, although this depends on microorganisms [14,30].

Table 8 shows that the isolated ACGs presented a higher lethality (*p* < 0.05) than crude extract in the two strains evaluated. On the other hand, the MIC values for *C. albicans* were 1.28 and 1.29 µg/mL using ACGs crude extract and isolated ACGs, respectively, while the MIC values for *C. tropicalis* were 0.04 and 0 05 µg/mL. 

There are no studies evaluating the lethality of ACGs extracts on microbial strains; however, the antimicrobial efficacy from aqueous extracts of *A. muricata* leaves has been demonstrated on *C. albicans* (15 mm of inhibition zone) using different concentrations [31].

The sublethal damage caused by the ACGs crude extract was 87% in *C. tropicalis* and 98% in *C. albicans.* The most significant sublethal damage occurred when *C. tropicalis* (90.4%) and *C. albicans* (98.6%) were treated with the isolated ACGs. Sublethal damage is described as any damage short of death. It is usually related to the high sensitivity of each microorganism to stress conditions after any treatment and its ability to survive adverse conditions. The injury may be metabolic or structural. Structural damage manifests as changes in the cell wall or cell membrane permeability, whereas metabolic damage is due to damage to the functional components of the cells. Both types of damage can result in a loss of growth capacity and an inability to form colonies on selective media under suitable conditions for the growth of intact cells. Conventionally, sub-damaged cells are defined as the population of cells that cannot grow in a minimal medium or medium containing a stressor as a selective agent but will grow in a nutrient-rich complete medium [32,33]. Rizqilah et al. [34] concluded that crude extracts from plants show less inhibition because the secondary metabolites can only interact with fungal cell membranes but do not diffuse into the cell; therefore, there is no interference with the nucleic acid.

Nonetheless, isolated or purified bioactive compounds cause microbial inhibition by adhesion of these compounds to the cell surface or by diffusion into the fungal cells. Antifungal substances in an extract can inactivate the function of the genetic material due to interfering with the formation of nucleic acids (DNA and RNA). Mendez-Chavez et al. [30] demonstrated that the methanolic extract from *Annona cherimola* leaves increased the fungal inhibition from 53.16% to 82.28% on *Fusarium oxysporum* strains; attributing this effect to the ACGs identified in this study (Annomolon-B, 34-epi annomolon B, almunequin, cherimoline 1, cherimoline 2, and isocherimoline 1).

## 3. Materials and Methods

### 3.1. Chemicals and Reagents

Two acetogenin samples (Annonacin and pseudoannonacin) from *A. muricata* pulp were provided by Zepeda-Vallejo’s research group (Department of Organic Chemistry of the National School of Biological Sciences, National Polytechnic Institute, Mexico City, Mexico). The chemical structure of the above acetogenins was established based on semipreparative-HPLC, HPLC–MS, and NMR analysis. The purity of each sample was determined to be above 95%. Silica gel, 2,2′-azino-bis (3-ethylbenzothiazoline-6-sulfonic acid; 2,2-diphenyl-1-picrylhydrazyl; methanol, acetonitrile, 3,5-hydroxybenzoic acid, 2,4,6-tripyridyl-s-triazine, 6-hydroxy-2,5,7,8-tetramethylchromane-2-carboxylic acid, ferric chloride hexahydrated and water-HPLC grade were purchased from Sigma-Aldrich Co. Ltd. (St. Louis, MO, USA). Petroleum ether, potassium hydroxide, dichloromethane, ethyl acetate, hexane, chloroform, ethanol, and acetone, all analytical grade, were purchased from Jalmek Scientific S.A., Guadalajara, Jalisco, Mexico.

### 3.2. Collection of Raw Material and Its Preparation

The ripened *A. muricata* fruits were harvested in an orchard in Compostela, Nayarit, Mexico (21°05′00.4″ N 105°08′50.8″ W). The seeds were obtained after manual depulping. They were dried in an oven (Memmert LL-50, Schwabach, Germany) for 24 h at 50 °C to 34.9 ± 1.14% of moisture. The tegument was eliminated from dried seeds, and the endosperms were pulverized in a high-speed multifunctional mill (CGoldenWall HC-2000, San Francisco, CA, USA) to 80–300 µm of particle size.

#### Endosperm Defatting

Endosperm defatting was performed using Soxhlet equipment (Novatech VH-6, Jalisco, Mexico) with 10 g of the sample placed in the extraction cartridge and 250 mL of petroleum ether. Defatting was carried out for 24 h, and the defatted endosperm was dried for 4 h at 50 °C [35].

### 3.3. Thermosonication-Assisted Extraction (TSAE) of Crude Acetogenins from the Defatted Endosperm of Annona muricata Seeds

A Box-Behnken design was used. The individual and interaction effects of sonication amplitude (XSA 80, 90, and 100%), pulse-cycle (XPC 0.5, 0.7, and 1 s), and sonication temperature (XTE 40, 50, and 60 °C) were evaluated. Temperatures were selected as a strategy to avoid solvent evaporation considering that the boiling temperature of methanol is 64.7 °C and they were stable (Appendix A); however, when there was a loss of dissolvent (only at 60 °C), methanol was added (2–3 mL) to complete total volume (35 mL). The range of the analyzed experimental conditions were based on experiments published by Aguilar-Hernández et al. [7], López-Ordaz et al. [11], and Ketenoglu [12]. TSAE was developed using a UP400S ultrasonic system (ultrasound power of 400 W, 24 kHz frequency) (Hielscher Ultrasonic, Teltow, Germany) with an ultrasonic probe (H7 Tip 7, Hielscher, Teltow, Germany) of maximum amplitude (175 µm) and an ultrasound intensity of 300 W/cm^2^. The defatted endosperm (2 g) was mixed with 35 mL of methanol previously heated at temperature according to design. The ultrasonic probe was immersed 2 cm in the methanolic solution, and extraction was by 50 min. The temperature was regulated with a cold-water recirculating bath (Thermo Scientific 2870, Waltham, MA, USA). The samples were centrifuged (Hermle Z32HK, Wehingen, Germany) at 11,624× *g* for 15 min at 4 °C. The residue was resuspended with methanol, and the extraction was repeated. The supernatants were joined and used for the ACGs analysis. The design was performed for three replicates.

#### 3.3.1. Total Acetogenins Content (TAC) and Yield

The TAC was determined according to the established procedure by Aguilar-Hernández et al. [7]. The colorimetric reaction was carried out with 50 µL of the methanolic extract and 2 mL of Kedde’s reagent (3,5-hydroxybenzoic acid dissolved in methanol and 5.7% potassium hydroxide solution). The absorbance was measured at 505 nm in a spectrophotometer (Jenway 6705, Dunmow, UK). The analysis was performed in triplicate. The TAC was calculated with a standard annonacin curve, and the results were expressed as milligram annonacin equivalents per gram of dry weight (mg/g DW). The yield percentage was calculated with Equation (3).
(3)$$\mathrm{Yield}\;\left(\%\right)=\frac{\mathrm{Total}\;\mathrm{acetogenins}\;\left(\mathrm{mg}\right)}{\mathrm{Sample}\;\left(g\right)}\times 100$$

### 3.4. Response Surface Analysis to Obtain the Optimal Tsae Conditions of Acetogenin Crude Extract from the Defatted Endosperm of Annona muricata Seeds

The optimal TSEA conditions for the acetogenin extraction and yield from the defatted endosperm of *A. muricata* seeds were obtained by applying the RSM. The predicted values were calculated from the second-order polynomial Equation (4).
(4)$$Y=\;\beta 0+\overset{Ε}{\underset{i=A}{\sum }}\beta i\;\mathrm{Xi}+\overset{Ε}{\underset{i=A}{\sum }}\overset{Ε}{\underset{j=A\neq i}{\sum }}\beta \mathrm{ij}\;\mathrm{Xi}+\;Ε$$
where Y is the predicted response (TAC or Yield), Xi is the coded or uncoded value for the factors (X_TE_, X_PC_, and X_AS_), β0 is a constant, βi is the main effect of the coefficient for each variable, and βij are the interaction effect coefficients. 

Model adequacy was evaluated by analysis of variance (ANOVA) to determine the significant effect of independent variables and correlation coefficients (square R) at a significance level of 5%. Moreover, data were adjusted (adjusted R) with the second-order polynomial equation by multiple regression using Statistic software (v. 10 Statsoft, Tulsa, OK, USA).

### 3.5. Model Reliability and Comparison of the Optimal TSAE Conditions with Ultrasound and Soxhlet Method to Obtain Acetogenin Crude Extract

The optimal TSAE conditions on TAC extraction were experimentally verified with three replicates to validate the model reliability. The antioxidant capacity (AOX) was also evaluated. Moreover, the results were compared with those obtained by UAE at 25 °C and the Soxhlet method.

UAE at 25 °C was performed using the same UP400S ultrasonic equipment (Hielscher Ultrasonics, Teltow, Germany) and ultrasonic probe. The procedure was started with 2 g of defatted endosperm sonicated with 35 mL of methanol at 100% amplitude, pulse-cycle of 0.5 s, and extraction time of 50 min at 25 ± 2 °C using a cold-water recirculating bath (Thermo Scientific 2870, Waltham, MA, USA). The extracts were centrifuged (11,624× *g*, 5 min, 4 °C). The residue was resuspended with methanol, and the extraction was repeated. The supernatants were joined and analyzed. 

The defatted endosperm (10 g) was weighed and placed in extraction cartridges, and 150 mL of methanol was added to the flask and placed in the Soxhlet equipment at 70 ± 2 °C for 10 h [26]. Then, the extract was concentrated in a rotary evaporator (Yamato RE300, Tokyo, Japan) for 20 mL and analyzed.

#### Total Acetogenin Content and Antioxidant Capacity

TAC was determined, as was mentioned in Section 3.3.1. The antioxidant capacity of the acetogenin extracts was evaluated by three in vitro methods. ABTS radical (2,2′-azino-bis-(3-ethylbenzothiazoline-6-sulfonic acid) assay was evaluated according to Re et al. [36] with some modifications. The samples (35 µL) were mixed with 265 µL of the ABTS (7 mM) and shaken in the dark for 7 min at 30 °C. The absorbance was measured at 734 nm. DPPH (1,1-Diphenyl-2-picrylhydrazyl) assay was performed based on the Prior et al. [37] method with some modifications. Briefly, the extract (40 µL) was reacted with 260 µL of DPPH solution (190 µM), and after 10 min in the dark, the absorbance was measured at 517 nm. Ferric reducing antioxidant power (FRAP) assay was carried out with the methodology described by Benzie and Strain [38]. The extracts (36 µL) were mixed with 9 µL of distilled water and 264 µL of FRAP reagent (solution 10:1:1 (*v*/*v*/*v*) of 0.3 M sodium acetate buffer, 10 mM 2,4,6-tripyridyl-s-triazine and 20 mM ferric chloride hexahydrated). After 30 min in the dark, the absorbance was measured at 595 nm. All absorbances were measured on a microplate reader (Bio-Tek Synergy HT, Winooski, VT, USA). Trolox (6-hydroxy-2,5,7,8-tetramethylchromane-2-carboxylic acid) was used as a standard for the calibration curve. The results were expressed in micromoles of Trolox equivalents per gram of dry weight (μmol TE/g DW). 

### 3.6. Solubility and Identification by HPLC-DAD of Isolated Acetogenins

#### 3.6.1. Isolation of Acetogenins by Column Chromatography

First, a methanolic extract (500 mL) was obtained under optimal TSAE conditions. The methanolic extract was evaporated to dryness in a rotavapor (Yamato RE300, Tokyo, Japan) to get a crude extract. Approximately 2 g of ACGs crude extract was subjected to a chromatographic column (6.4 × 57.0 cm). Silica gel (60 mesh) was used as a stationary phase, and dichloromethane: ethanol mixture (9:1 *v*/*v*) was used as an initial eluent with a gradual increase in polarity until finishing with 100% ethanol. The obtained fractions were subjected to thin-layer chromatography (TLC) and revealed with Kedde reagent to confirm the presence or absence of ACGs [7]. The ACG-rich fractions were joined (F1). F1 was subjected to a flash chromatography column (6 cm × 25 cm, SiO_2_, 230–400 mesh), using a mixture of dichloromethane: ethyl acetate as eluent. The ratio of eluents during the elution was 98:2 (*v*/*v*) until ACGs stopped eluting (91:9 *v*/*v*). The fractions were submitted to TLC. Those rich in ACGs were joined (F2) and subjected to an additional assay by flash column chromatography for the final isolation process. The resultant fractions were submitted to TLC to select four ACG-positive fractions, and they were joined (F3). Finally, F3 was concentrated to dryness and used to evaluate its solubility to be analyzed by HPLC-DAD. 

#### 3.6.2. Solubility of Isolated Acetogenins

Ten solvents (water, petroleum ether, acetonitrile, hexane, dichloromethane, chloroform, ethyl acetate, ethanol, methanol, and acetone) were used for the solubility test. Briefly, 1 mg isolated acetogenins (F3) was dissolved in 1 mL of each solvent. Subsequently, the mixture was shaken for 1 min in a vortex until complete homogenization. The test was repeated until the saturation of ACGs in each solvent. The results were reported in dissolved milligrams per milliliter of solvent (mg/mL).

#### 3.6.3. Analysis of Acetogenins by HPLC-DAD

Partial identification of acetogenins was carried out according to the conditions established by Yang et al. [5,6]. Isolated acetogenins (F3) were resuspended in methanol (0.22 mg/mL) and, before injection, were filtered through 0.22 µm membrane filters. Samples (10 µL) were injected into an HPLC system (Agilent Technologies 1260 Infinity, Waldbronn, Germany) equipped with a photodiode array detector (DAD) and an Agilent Zorbax Extend C18 reverse-phase column (250 mm × 4.6 mm, 5 µm) at 30 °C. The mobile phase consisted of methanol (eluent A) and water (eluent B), using a linear gradient: 0–40 min (85% A) and 40–60 min (85–95% A) at a flow rate was 1 mL/min. In addition, other separation tests were made by changing the column temperature to 10 °C, 15 °C, 20 °C, or 25 °C. ACGs were detected at 220 nm. 

Subsequently, the identification of ACGs was analyzed according to the chromatographic results presented by Yang et al. [5,6]. The results are shown as area values and percentages of each ACG. In addition, two acetogenins were quantified with calibration curves.

### 3.7. Antifungal Activity of Crude Extract and Isolated Acetogenins

The antifungal activity of the crude extract and isolated acetogenins on *Candida albicans* (ATCC), *C. glabrata* (ATCC 15126), *C. tropicals* (ATCC 1369), and *C. krusei* (ATCC 14243) strains was carried out by the disk diffusion method according to Anaya-Esparza et al. [39] with some modifications. The strains were grown aerobically in nutrient broth (8 g/L, pH 7.0 ± 0.1) for 24 h at 37 °C until the suspensions reached 1 × 10^6^ CFU/mL compared to 0.5 of the McFarland standard. The disk diffusion assay was performed by passing each *Candida* strain through nutrient agar (23 g/L, pH 6.8 ± 0.1). Subsequently, sterile filter paper discs of 7 mm diameter were impregnated (200 μL) with suspensions of the crude extract and isolated ACGs dissolved in dimethyl sulfoxide (DMSO) at concentrations of 12.5, 25, 50, 100, 200, 400, 800, 1000, 2000, and 4000 µg/mL and placed in inoculated Petri dishes using sterile forceps. The antibiotic ketoconazole (500 µg/mL) was used as a positive control, sterile distilled as the negative control, and DMSO as solvent. Subsequently, the Petri dishes were incubated at 37 °C for 48 h. The diameter of the inhibition zone formed around the disks was measured with a Vernier and reported in millimeters (mm). The procedure described above was performed in triplicate for each concentration and each *Candida* strain in both extracts.

According to the highest inhibition (mm) presented by the ACGs crude extract and isolated ACGs in the different strains, two strains (*C. albicans* and *C. tropicalis*) were selected to calculate the medium inhibitory concentration (IC_50_). IC_50_ was used to carry out the sublethal damage and lethality assays in the same chosen strains. Moreover, the minimum inhibitory concentration (MIC) of crude extract and isolated ACGs in these strains was calculated.

Lethality and sublethal damage of crude extract and isolate ACGs on *C. albicans,* and *C. tropicalis* strains were evaluated by serial dilution assay using the pour-plate method established by Anaya-Esparza et al. [32]. The nutrient broth (200 mL) was inoculated with 10 mL/L of cell suspension (10^6^ CFU/mL) adjusted according to McFarland’s scale (0.5 of absorbance), spiked with the ACGs (100 µg/mL), and incubated for 15 min at 37 °C. Subsequently, 1 mL of the above solution was transferred to 9 mL of sterile saline (0.85% *w*/*v*) and homogenized. Serial dilutions (1 × 10^−1^ to 1 × 10^−7^ CFU/mL) were made in saline (9 mL), and 1 mL aliquots were taken and seeded on nutrient agar, and plates were poured. This procedure was repeated for each extract and with each strain, and the results were reported as log CFU/mL. Lethality was calculated as the difference between the logarithms of the colony counts in the control group without ACGs (N_0_) and the colony counts in the samples treated with the extracts (N) (Log CFU/mL N_0_-Log CFU/mL N). 

The sublethal damage was calculated as the difference between the CFU of the control group and the CFU of strains treated with ACGs and expressed as a percentage (%).

### 3.8. Statistical Analysis

Data were expressed as means ± standard deviation (*n* = 6). The results from the Box–Behnken design were analyzed with the RSM. Subsequently, the other experiments were conducted in a one-factorial experimental design. The data were performed using ANOVA (*p* < 0.05) with Statistic software (v.10 Statsoft, Tulsa, OK, USA). Tukey test examined the mean differences between treatments (α = 0.05).

## 4. Conclusions

To the best of our knowledge, it is the first study using thermosonication as an effective extraction tool of Annonaceous acetogenins. The optimal TSAE conditions were 100% sonication amplitude, 0.5 s pulse-cycle, and 50 °C. In addition, TSAE was more efficient in extracting ACGs from *A. muricata* seeds than UAE at 25 °C and the Soxhlet method. Isolated ACGs were highly soluble in acetone, methanol, and ethanol but insoluble in water, acetonitrile, and non-polar solvents. The presence of pseudoannonacin and annonacin was demonstrated, and eight additional acetogenins. It is the first time that pseudoannonacin has been identified in *A. muricata* seeds in great abundance. The crude and isolated ACGs (100 µg/mL) showed the highest antifungal effect against *C. albicans* and *C. tropicalis*. The present work demonstrated that TSAE is an advantageous method for extracting ACGs from the defatted endosperm of *A. muricata* seeds. TSAE is an alternative to implement the large-scale extraction of ACGs. Moreover, once isolated, ACGs can be used to formulate therapeutic drugs and antifungal treatments.

## Figures and Tables

**Figure 1 molecules-27-06045-f001:**
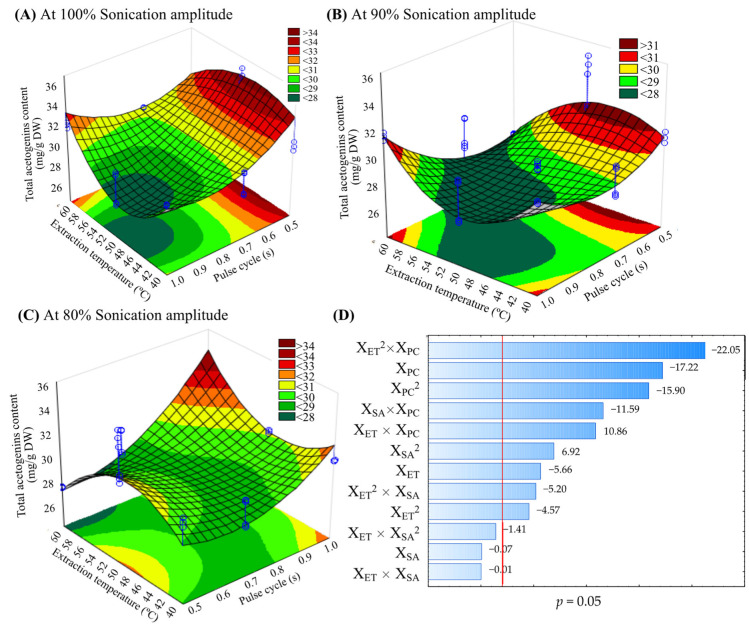
Response surface plots indicating the effect of thermosonication-assisted extraction on the total acetogenin content from the defatted endosperm of *A. muricata* seeds using 100% (**A**), 90% (**B**) and 80% (**C**) of sonication amplitude and Pareto plot (**D**). DW = dry weight; X_ET_ = Extraction temperature (°C); X_SA_ = Sonication amplitude (%); X_PC_ = Pulse-cycle (s).

**Figure 2 molecules-27-06045-f002:**
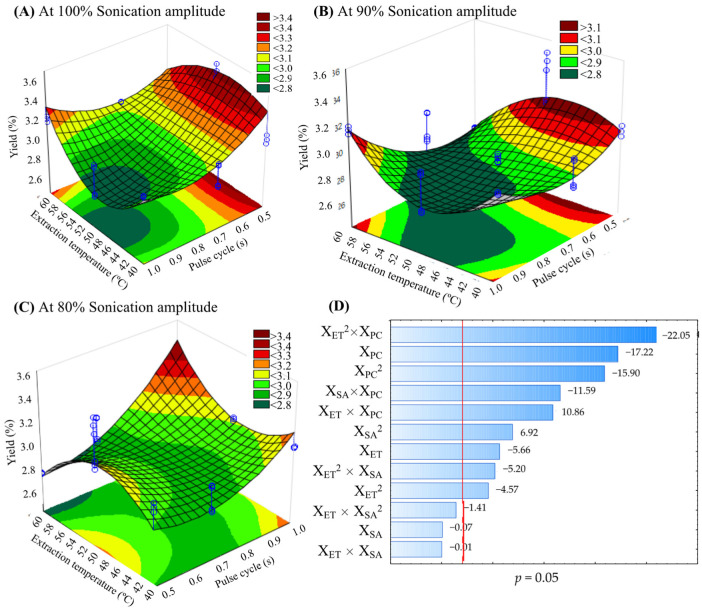
Response surface plots indicating the effect of thermosonication-assisted extraction on yield (%) of acetogenins from the defatted endosperm of *A. muricata* seeds using 100% (**A**), 90% (**B**) and 80% (**C**) of sonication amplitude and Pareto plot (**D**). DW = dry weight; X_ET_ = Extraction temperature (°C); X_SA_ = Sonication amplitude (%); X_PC_ = Pulse-cycle (s).

**Figure 3 molecules-27-06045-f003:**
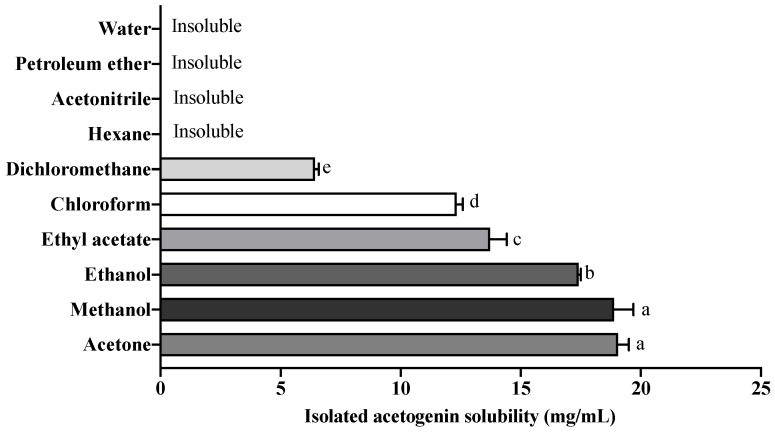
Solubility of isolated acetogenins in different solvents. Different letters indicate significant statistical differences (α = 0.05).

**Figure 4 molecules-27-06045-f004:**
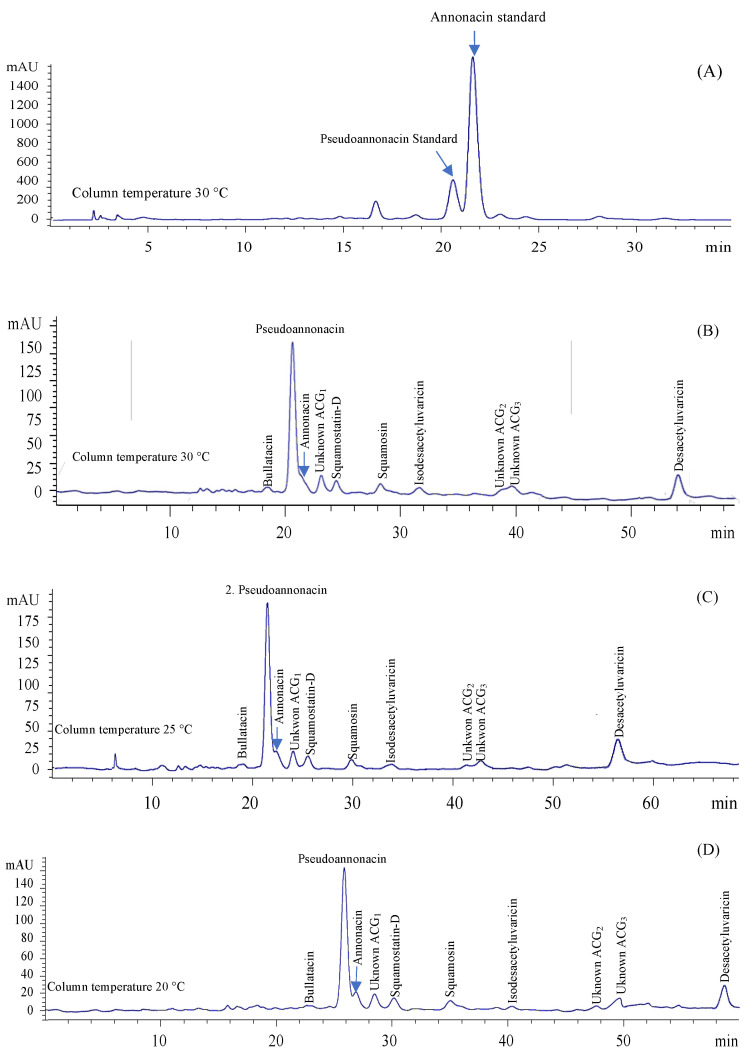
Representative HPLC-chromatograms of acetogenins standards (**A**) and isolated acetogenins from *A. muricata* seeds when column temperature was maintained at 30 °C (**B**), 25 °C (**C**), and 20 °C (**D**).

**Figure 5 molecules-27-06045-f005:**
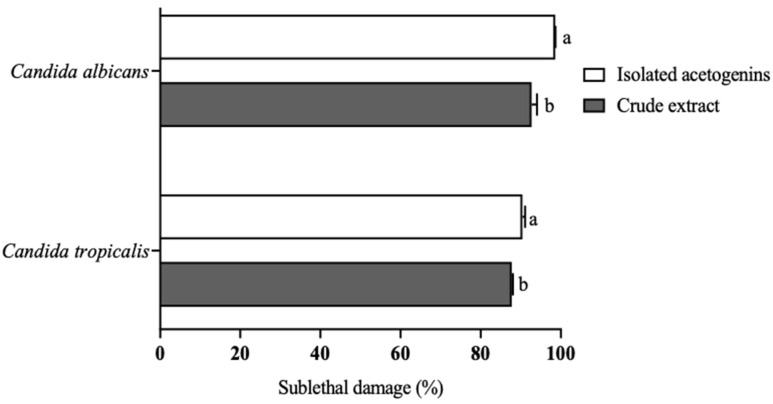
Sublethal damage (%) of crude extract and isolated acetogenins (100 µg/mL) against *Candida albicans* and *Candida tropicalis*. Different letters indicate significant statistical differences between treatments by each microorganism (α = 0.05).

**Table 1 molecules-27-06045-t001:** Experimental values of total acetogenins and yield after thermosonication-assisted extraction (TSAE) from the defatted endosperm of *A. muricata* seeds; also predicted values and error rates after the response surface analysis.

Run	TSAE Conditions			Total Acetogenins (mg/g DW)			Yield (%)		
	X_ET_ (°C ± 2)	X_SA_ (%)	X_PC_ (s)	Experimental	^1^ Predicted	Error Rate (%)	Experimental	^1^ Predicted	Error Rate (%)
1	40	80	0.7	28.33 ± 0.11 ^ef^	29.17	−0.84	2.83 ± 0.01 ^ef^	2.91	−0.08
2	60	80	0.7	28.22 ± 0.21 ^efg^	28.49	−0.26	2.82 ± 0.02 ^fg^	2.84	−0.02
3	40	100	0.7	30.38 ± 0.09 ^cd^	30.11	0.27	3.04 ± 0.01 ^cd^	3.01	0.03
4	60	100	0.7	30.27 ± 0.03 ^cd^	29.43	0.84	3.03 ± 0.01 ^cg^	2.94	0.09
5	40	90	0.5	30.74 ± 0.40 ^bcd^	30.83	−0.09	3.07 ± 0.04 ^bcd^	3.08	−0.01
6	60	90	0.5	27.40 ± 0.07 ^gh^	30.16	−2.76	2.74 ± 0.01 ^gh^	3.01	−0.27
7	40	90	1	30.64 ± 0.09 ^cd^	29.98	0.66	3.06 ± 0.01 ^cd^	2.99	0.07
8	60	90	1	31.50 ± 0.29 ^b^	29.31	2.19	3.15 ± 0.03 ^b^	2.93	6.08
9	50	80	0.5	31.11 ± 0.32 ^bc^	30.51	0.60	3.11 ± 0.03 ^bc^	3.05	0.06
10	50	100	0.5	34.35 ± 0.68 ^a^	31.78	2.24	3.44 ± 0.07 ^a^	3.17	0.27
11	50	80	1	30.15 ± 0.15 ^d^	29.66	0.50	3.02 ± 0.01 ^d^	2.96	0.06
12	50	100	1	27.26 ± 0.12 ^h^	30.60	−3.35	2.73 ± 0.01 ^h^	3.06	−0.33
13	50	90	0.7	28.30± 0.41 ^ef^	28.20	0.10	2.83 ± 0.04 ^ef^	2.82	0.01
14	50	90	0.7	28.59 ± 0.32 ^e^	28.20	0.39	2.86 ± 0.03 ^e^	2.82	0.04
15	50	90	0.7	27.70 ± 0.06 ^fgh^	28.20	−0.50	2.77 ± 0.01 ^fgh^	2.82	−0.05

TSAE = Thermosonication-assisted extraction; X_ET_ = extraction temperature; X_SA_ = sonication amplitude; X_PC_ = pulse-cycle. Data are expressed as means ± standard deviation of three determinations and three replicates (*n* = 9). Different letters by column indicate significant statistical differences between treatments (α = 0.05). ^1^ The values were predicted using the secondary polynomial equations, R^2^ = 0.97.

**Table 2 molecules-27-06045-t002:** Predicted mathematical models for extracting acetogenins and yield from *Annona muricata* seeds after thermosonication-assisted extraction.

Response	Proposed Model	Equation
Total acetogenins (mg/g)	−548.77 − 0.14X_ET_^2^ + 6.96X_SA_ − 0.03X_SA_^2^ + 29.13X_PC_^2^ + 0.001X_ET_ × X_SA_^2^ + 0.001X_ET_^2^ × X_SA_μ11.63X_ET_ × X_PC_ + 0.12X_ET_^2^*X_PC_ − 0.61X_SA_ × X_PC_ + 422.74	(1)
Yield (%)	−54.88 − 0.01 × X_ET_^2^ + 0.70X_SA_ − 0.003X_SA_^2^ + 2.91X_PC_^2^ + 0.0001X_ET_ × X_SA_^2^ + 0.0001X_ET_^2^ × X_SA_ − 1.16X_ET_ × X_PC_ + 0.01X_ET_^2^ × X_PC_ − 0.06X_SA_ × X_PC_ + 42.27	(2)

The individual effects of each independent variable and their interactions on the TAC and yield are X_ET_ = Extraction temperature (°C); X_SA_ = Sonication amplitude (%); X_PC_ = Pulse-cycle (s).

**Table 3 molecules-27-06045-t003:** Analysis of variance and regression coefficients of predicted quadratic polynomial models with the thermosonication-assisted extraction conditions on total acetogenins (TAC) and yield from the defatted endosperm of *A. muricata* seeds.

Source	Analysis of Variance					Regression Coefficients
	SS	DF	MS	F Value	*p*-Value	
TAC						β-coefficient
Mean/intercept						−548.770 *
X_ET_^2^	3.0319	1	3.03194	27.0829	0.000011 *	−0.142 *
X_SA_	5.3620	1	5.36199	47.8962	0.000001 *	6.956 *
X_SA_^2^	15.038	1	15.0380	134.327	0.000001 *	−0.027 *
X_PC_^2^	33.216	1	33.2168	296.710	0.000001 *	29.127 *
X_ET_ × X_SA_^2^	3.5864	1	3.58645	32.0361	0.000003 *	0.001 *
X_ET_^2^ × X_SA_	2.3337	1	2.33368	20.8457	0.000070 *	0.001 *
X_ET_ × X_PC_	13.200	1	13.2007	117.916	0.000001 *	−11.631 *
X_ET_^2^ × X_PC_	54.454	1	54.4541	486.414	0.000001 *	0.121 *
X_SA_ × X_PC_	28.306	1	28.3067	252.851	0.000001 *	−0.614 *
Lack of Fit	0.2224	3	0.07414	0.66230	0.581338 **	
Pure Error	3.5824	32	0.11			
R-square	0.9775					
R-adjust	0.9691					
Total SS	159.34					
**Yield (%)**						β-coefficient
Mean/intercept						−54.8770 *
X_ET_^2^	0.0303	1	0.030319	27.0829	0.000011 *	−0.0142 *
X_SA_	0.0536	1	0.053620	47.8962	0.000001 *	0.6956 *
X_SA_^2^	0.1504	1	0.150380	134.3277	0.000001 *	−0.0027 *
X_PC_^2^	0.33217	1	0.332168	296.7109	0.000001 *	2.9127 *
X_ET_ × X_SA_^2^	0.03586	1	0.035864	32.0361	0.000003 *	0.0001 *
X_ET_^2^ × X_SA_	0.02334	1	0.023337	20.8457	0.000070 *	0.0001 *
X_ET_ × X_PC_	0.13201	1	0.132007	117.9163	0.000001 *	−1.1631 *
X_ET_^2^ ×X_PC_	0.54454	1	0.544542	486.4142	0.000001 *	0.0121 *
X_SA_ × X_PC_	0.28307	1	0.283068	252.8514	0.000001 *	−0.0614 *
Lack of Fit	0.002224	3	0.000741	0.6623	0.581338 **	
Pure Error	0.035824	32	0.001120			
R-square	0.9775					
R-adjust	0.9691					
Total SS	1.5935					

X_ET_ = Extraction temperature; X_SA_ = sonication amplitude; X_PC_ = pulse cycle; SS = Sum of square; DF = Degree of freedom; MS = Means square. * Significant (*p* < 0.05); ** nonsignificant (*p* > 0.05).

**Table 4 molecules-27-06045-t004:** Optimal conditions by thermosonication-assisted extraction of total acetogenins and yield from the defatted endosperm of *A. muricata* seeds obtained by the predicted models.

Parameter	Total Acetogenins (mg/g)	Yield (%)
Extraction temperature (°C)	50	50
Sonication amplitude (%)	100	100
Pulse cycle (s)	0.5	0.5
Extraction time (min)	50	50
Optimal response	33.98	3.43
−95% Confidence limit	32.22	3.36
+95% Confidence limit	35.78	3.51

Confidence limits of −95%, lower limit; +95% confidence limit, upper limit; confidence interval are the difference between upper and lower limits.

**Table 5 molecules-27-06045-t005:** Total acetogenins and antioxidant capacity of extracts obtained from the defatted endosperm of *A. muricata* seeds using optimal extraction conditions assisted by thermosonication (TSAE), ultrasound-assisted extraction (UAE), and soxhlet method.

Parameter	^1^ TSAE	^2^ UAE	^3^ Soxhlet
Total acetogenins (mg/g DW)	35.89 ± 0.59 ^a^	16.52 ± 1.87 ^b^	2.30 ± 0.12 ^c^
Yield (%)	3.6	1.65	0.23
TSAE effectiveness (*n*-fold)		2.17	15.60
Antioxidant capacity (μmol/g DW)			
ABTS	4308.09 ± 89.49 ^a^	3336.06 ± 56.61 ^b^	1067.18 ± 73.91 ^c^
DPPH	1668.29 ±18.22 ^a^	1315.08 ± 118.98 ^b^	450.00 ± 49.51 ^c^
FRAP	1512.89 ± 24.38 ^a^	1334.17 ± 89.17 ^b^	694.87 ± 254.76 ^c^

DW = dry weight; ABTS = 2,2′-azino-bis (3-ethylbenzothiazoline-6-sulfonic acid; DPPH = 2,2-diphenyl-1-picrylhydrazyl; FRAP = Ferric ion reducing antioxidant power. All values are mean ± standard deviation of three determinations and three replicates (*n* = 9). Different letters in each file indicate significant statistical differences between treatments (α = 0.05).^1^ 100% X_SA_, 0.5 s X_PC_, 50 ± 2 °C and 50 min. ^2^ 100% X_SA_, 0.5 s X_PC_, 25 ± 2 °C and 50 min. ^3^ 70 ± 2 °C and 10 h.

**Table 6 molecules-27-06045-t006:** Isolated acetogenins from the defatted endosperm of *A. muricata* seeds using the optimal extraction conditions assisted by thermosonication.

No. Peak	Acetogenin	Formula	MW	RT	^1^ Area Values	^1^ Area (%)	Reference
1	Bullatacin	C_37_H_66_O_7_	622.9	19.073	226.20 ± 11.90 ^g^	2.16	Yang et al. [5,6]
2	Pseudoannonacin	C_35_H_67_O_7_	596.9	21.223	6300.6 ± 12.00 ^a^	60.22	Identified with a standard
3	Annonacin	C_35_H_64_O_7_	596.9	22.125	645.05 ± 2.30 ^c^	6.17	Identified with standard
4	Unknown ACG_1_	-	-	23.706	636.95 ± 32.90 ^c^	6.09	-
5	Squamostatin-D	C_37_H_66_O_7_	622.9	25.049	456.25 ± 9.10 ^d^	4.36	Yang et al. [5,6]
6	Squamocin	C_37_H_66_O_7_	622.9	28.990	312.35 ± 9.30 ^f^	2.99	Yang et al. [5,6]
7	Isodesacetyluvaricin	C_37_H_66_O_6_	606.9	32.548	360.50 ± 5.20 ^e^	3.45	Yang et al. [5,6]
8	Unknown ACG_2_	-	-	38.325	323.5 ± 6.8 ^ef^	3.09	-
9	Unknown ACG_3_	-	-	39.227	352.8 ± 9.6 ^e^	3.37	-
10	Desacetyluvaricin	C_37_H_66_O_6_	606.9	54.719	848.35 ± 2.1 ^b^	8.10	Yang et al. [5,6]
Total					10462.55 ± 101.2	100	-

MW = Molecular weight (g/mol). RT = Retention time. ^1^ Arbitrary units. ACG = acetogenin. Different letters indicate significant statistical differences between ACGs areas (α = 0.05).

**Table 7 molecules-27-06045-t007:** Antifungal activity of ACGs crude extract and isolated ACGs from *A. muricata* seed on *Candida albicans, Candida krusei, Candida tropicalis,* and *Candida glabrata*.

Concentration (µg/mL)	Inhibition Zone (mm)							
	*C. albicans*		*C. krusei*		*C. tropicalis*		*C. glabrata*	
Ketoconazole (C+)	14.75 ± 0.5		NI		NI		36.25 ± 0.01	
Distilled water (C−)	NI		NI		NI		NI	
Dimethyl sulfoxide	NI		NI		NI		NI	
	C-extract	I-ACGs	C-extract	I-ACGs	C-extract	I-ACGs	C-extract	I-ACGs
800	12.00 ± 0.01 ^aY^	15.50 ± 0.58 ^aX^	12.00 ± 0.01 ^bY^	13.50 ± 0.58 ^aX^	12.50 ± 0.01 ^aY^	14.00 ± 0.58 ^aX^	8.25 ± 0.01 ^bX^	8.50 ± 0.58 ^aX^
400	11.25 ± 0.50 ^bY^	14.75 ± 1.71 ^abX^	12.75 ± 0.50 ^aY^	14.00 ± 0.01 ^aX^	13.00 ± 0.50 ^aX^	12.75 ± 1.71 ^cX^	8.00 ± 0.50 ^bX^	9.25 ± 0.01 ^aX^
200	10.75 ± 0.50 ^bY^	14.25 ± 1.50 ^abX^	12.50 ± 0.58 ^abX^	12.25 ± 0.50 ^bX^	12.50 ± 0.50 ^aX^	13.25 ± 1.50 ^bX^	8.75 ± 0.58 ^aX^	9.25 ± 0.50 ^aX^
100	8.00 ± 0.10 ^cY^	12.25 ± 0.50 ^cX^	12.00 ± 0.01 ^bY^	12.25 ± 0.50 ^bX^	11.50 ± 0.10 ^bX^	12.00 ± 0.50 ^dX^	8.50 ± 0.01 ^abX^	9.00 ± 0.50 ^aX^
50	8.00 ± 0.10 ^cY^	9.25 ± 0.50 ^dX^	11.75 ± 0.50 ^bX^	12.25 ± 0.50 ^bX^	11.75 ± 0.10 ^bX^	12.00 ± 0.50 ^dX^	8.00 ± 0.50 ^bX^	8.75 ± 0.50 ^aX^
25	8.00 ± 0.10 ^cX^	8.50 ± 0.58 ^eX^	11.00 ± 0.00 ^cX^	11.50 ± 0.58 ^cX^	11.00 ± 0.10 ^cX^	12.00 ± 0.58 ^dX^	8.00 ± 0.01 ^bX^	8.75 ± 0.58 ^aX^
12.5	8.00 ± 0.10 ^cX^	8.50 ± 0.58 ^eX^	10.25 ± 0.50 ^dX^	11.50 ± 0.58 ^cX^	11.00 ± 0.10 ^cX^	12.00 ± 0.58 ^dX^	8.00 ± 0.50 ^bX^	8.75 ± 0.58 ^aX^

Data are expressed as means of triplicate determinations ± standard deviation (*n* = 3). Different lowercase letters (a, b, c, d, e) in the same column indicate significant statistical differences between concentrations (*α* = 0.05). Different capital letters (X, Y) indicate significant statistical differences between ACGs treatments (*α* = 0.05). C+ = Positive control (500 µg/mL). C− = Negative control (Distilled water). NI = Not inhibition. C-extract = ACGs crude extract. I-ACGs = isolated ACGs.

**Table 8 molecules-27-06045-t008:** Lethality and minimum inhibitory concentration (MIC) of crude extract and isolated acetogenins on *C. albicans* and C. *tropicalis*.

Concentration (100 µg/mL)	Lethality (log CFU/mL)		MIC (µg/mL)	
	*C. albicans*	*C. tropicalis*	*C. albicans*	*C. tropicalis*
Crude extract	1.14 ± 0.12 ^b^	0.91 ± 0.02 ^a^	1.29 ± 0.06 ^a^	0.055 ± 0.01 ^a^
Isolated acetogenins	1.87 ± 0.02 ^a^	1.02 ± 0.04 ^a^	1.28 ± 0.01 ^a^	0.04 ± 0.01 ^a^

Values are the average ± standard deviation (*n* = 3). Different letters in each column indicate significant statistical differences between treatments (α = 0.05).

## Data Availability

The dataset used and/or analyzed the current study are available from the corresponding author on reasonable request.

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
