# Peer review of "Extraction of Acetogenins Using Thermosonication-Assisted Extraction from Annona muricata Seeds and Their Antifungal Activity"

_molecules, 2022, doi:10.3390/molecules27186045_

Round 1

Reviewer 1 Report

The research proposal adopted in the manuscript is well founded. The methodologies used and the conduction of the experiments are adequate. Data presentation and statistical analysis were well performed. The manuscript must be reviewed in the following items:

1-) provide geographic coordinates for sample collection;

2-) Reference for choosing the range of the analyzed experimental conditions;

3-) Authors must present the model equation considering only the significant terms.

Author Response

Thank you very much for your accurate comments. We have done our best to follow up all your recommendations. Note: Changes in the revised manuscript are highlighted in red letters.

Reviewer 1

1-) provide geographic coordinates for sample collection;

Line 388. It was added

2-) Reference for choosing the range of the analyzed experimental conditions;

Lines 406-407. It was added

3-) Authors must present the model equation considering only the significant terms.

It was corrected

Reviewer 2 Report

The manuscript of López-Romero et al. reports the thermosonication-assisted extraction from Annona muricata seeds and their antifungal activity.

Although data on the extraction from different parts of A. muricata fruit have already been published (10.3390/molecules25051139), the current manuscript goes into greater depth on the ultrasound extraction process from the fruit seeds.

The manuscript is well organized and written. However, some important points need to be better discussed in the manuscript, as follows:

1)        The authors should report how they avoid the methanol phase transition during the 50 min extraction time at 60 ºC

2)        The temperature profile of the extraction medium should be provided as supplementary material since it is well known that the energy input from ultrasound changes the temperature of the extraction medium even under using a recirculating heat-water bath

3)        Also, the authors should explain how they controlled the extraction temperature in the UAE at 25 ºC. On this point, it is unfair to compare the TSAE with UAE since it is largely reported that temperature plays an essential role in solid-liquid extraction. In my opinion, it is more important to access the effect of US waves in the extraction by comparing the processes at the same temperature but with and without US application

4)        The authors also should explain why they chose methanol as the extraction solvent since the solubility of isolated acetogenins was similar to these solvents, and it is very important nowadays to use green solvents as ethanol

5)        Lines 115-117: if the increasing temperature in the ultrasonic medium decreases the implosion of the bubbles, how can the authors affirm that a significant number of bubbles will implode in a short period?

It is necessary to discuss the effect of temperature and cavitation and the interaction of these variables. Moreover, it is preferable to discuss such mechanism extraction instead of performing process optimization by RSM.

Author Response

Thank you very much for your accurate comments. We have done our best to follow up all your recommendations. Note: Changes in the revised manuscript are highlighted in red letters.

Reviewer 2

The manuscript is well organized and written. However, some important points need to be better discussed in the manuscript, as follows:

1) The authors should report how they avoid the methanol phase transition during the 50 min extraction time at 60 ºC

It was added in lines 402-405

2) The temperature profile of the extraction medium should be provided as supplementary material since it is well known that the energy input from ultrasound changes the temperature of the extraction medium even under using a recirculating heat-water bath.

The reviewer is right in the change of temperature; however due to this, we controlled the temperature of manner correct with a cold-water recirculating bath, no with a heat-water bath, it was our mistake in the document.  We include the temperature changes in the supplementary material.

3) Also, the authors should explain how they controlled the extraction temperature in the UAE at 25 ºC. On this point, it is unfair to compare the TSAE with UAE since it is largely reported that temperature plays an essential role in solid-liquid extraction. In my opinion, it is more important to access the effect of US waves in the extraction by comparing the processes at the same temperature but with and without US application.

The temperature was controlled with a cold-water recirculating bath (see lines 448-449) it is mean that the temperature was constant 25 °C ±1. And respect to this it is more important to access the effect of US waves in the extraction by comparing the processes at the same temperature but with and without US application”. We extracted ACGs applying only temperature (Soxhlet method) and in the same sense only cavitation by ultrasonic waves (US) controlling the temperature at 25 °C because the main objective of this work was using thermosonication (the combination of temperature and heat) and to corroborate an additive or synergistic effect.

4) The authors also should explain why they chose methanol as the extraction solvent since the solubility of isolated acetogenins was similar to these solvents, and it is very important nowadays to use green solvents as ethanol.

The reviewer is right, the best is to extract ACGs with ethanol; the explication is because before several tests were carried out in the laboratory with different dissolvents concluding that methanol increases the ACGs extraction (Aguilar-Hernández et al. 2022, Biotecnia, 25(2): 12-19), but we did not include ethanol, due to many references recommended principally methanol or chloroform as dissolvents to extract ACGs. Therefore, we decided to use methanol as extraction solvent of ACGS according to references and ourself screening. In this work we discovered that isolated ACGs are highly soluble in ethanol in the solubility test. Until now there was not a report of solubility of isolated ACGs; therefore, this result has as a perspective in future investigations to extract ACGs with ethanol (green dissolvent).

5) Lines 115-117: if the increasing temperature in the ultrasonic medium decreases the implosion of the bubbles, how can the authors affirm that a significant number of bubbles will implode in a short period?. It is necessary to discuss the effect of temperature and cavitation and the interaction of these variables. Moreover, it is preferable to discuss such mechanism extraction instead of performing process optimization by RSM

We believe that there is some confusion regarding this observation, since in line 117-129 it is explained that “the increase of temperature in the ultrasonic medium decreases the threshold of bubble implosion in cavitation. Thus, a larger number of bubbles will implode in a short period of time causing considerable damage to the cell wall or an increase in cell pore size. It improves the diffusivity of the solvent within the matrix, increasing the yield; however, the effect depends on the TSAE conditions (Mason et al., 1996; Anaya-Esparza et al., 2017). In this study, the increase in the yield of ACGs is attributed to the previous preparation of the raw material (defatted) and the synergistic effect between temperature (50 °C), sonication amplitude (100%), and pulsed sonication (0.5 s pulse-cycle). Nonetheless, if the extraction temperature is 40 ± 2 °C, it is insufficient to increase the yield independently of sonication amplitude and pulse-cycle. However, if the extraction temperature is 60 ± 2 °C, 100% sonication amplitude, and 1 s pulse-cycle, there is an excess of energy in the medium, which degrades ACGs [17]. According to Neske et al. [18], the structures of ACGs are liable to change above 60 °C”.

Reviewer 3 Report

The comments are as follows:

1. Please, explain the abbreviation of ACG in the abstract section.

2. The overall, more concrete and realistic conclusion is missing in the abstract section.

3. More details about plant material preparation (particle size, moisture content) are necessary.

4.  Please, improve the conclusion section with future perspective.

Author Response

Thank you very much for your accurate comments. We have done our best to follow up all your recommendations. Note: Changes in the revised manuscript are highlighted in red letters.

Reviewer 3

  1. Please, explain the abbreviation of ACG in the abstract section.

It was corrected. Line 26

  1. The overall, more concrete and realistic conclusion is missing in the abstract section.

It was added. Line 35-37

  1. More details about plant material preparation (particle size, moisture content) are necessary.

It was added. Lines 390 and 392.

  1. Please, improve the conclusion section with future perspective.

It was added. Line 569-571

Round 2

Reviewer 2 Report

I recommend the publication of the manuscript in the present form. Congratulations to the authors!